# Dietary supplementations to mitigate the cardiopulmonary effects of air pollution toxicity: A systematic review of clinical trials

Mehran Ilaghi[1], Fatemeh Kafi[2], Mohadeseh Shafiei[3], Moein Zangiabadian[4]*, Mohammad Javad Nasiri[5]*

1 Institute of Neuropharmacology, Kerman Neuroscience Research Center, Kerman University of Medical Sciences, Kerman, Iran, 2 School of Medicine, Isfahan University of Medical Sciences, Isfahan, Iran, 3 Faculty of Medicine, Kerman University of Medical Sciences, Kerman, Iran, 4 Endocrinology and Metabolism Research Center, Institute of Basic and Clinical Physiology Sciences, Kerman University of Medical Sciences, Kerman, Iran, 5 Department of Microbiology, School of Medicine, Shahid Beheshti University of Medical Sciences, Tehran, Iran

* zangiabadian1998@gmail.com (MZ); mj.nasiri@hotmail.com (MJN)

**Data Availability Statement:** All relevant data are within the paper and its Supporting Information files.

## Abstract

### Background

There is a consistent association between exposure to air pollution and elevated rates of cardiopulmonary illnesses. As public health activities emphasize the paramount need to reduce exposure, it is crucial to examine strategies like the antioxidant diet that could potentially protect individuals who are unavoidably exposed.

### Methods

A systematic search was performed in PubMed/Medline, EMBASE, CENTRAL, and ClinicalTrials.gov up to March 31, 2023, for clinical trials assessing dietary supplements against cardiovascular (blood pressure, heart rate, heart rate variability, brachial artery diameter, flow-mediated dilation, and lipid profile) or pulmonary outcomes (pulmonary function and airway inflammation) attributed to air pollution exposure.

### Results

After reviewing 4681 records, 18 studies were included. There were contradictory findings on the effects of fish oil and olive oil supplementations on cardiovascular outcomes. Although with limited evidence, fish oil offered protection against pulmonary dysfunction induced by pollutants. Most studies on vitamin C did not find protective cardiovascular effects; however, the combination of vitamin C and E offered protective effects against pulmonary dysfunction but showed conflicting results for cardiovascular outcomes. Other supplements like sulforaphane, L-arginine, n-acetylcysteine, and B vitamins showed potential beneficial effects but need further research due to the limited number of existing trials.

**Funding:** The author(s) received no specific funding for this work.

**Competing interests:** The authors have declared that no competing interests exist.

## Conclusions

Although more research is needed to determine the efficacy and optimal dose of anti-inflammatory and antioxidant dietary supplements against air pollution toxicity, this low-cost preventative strategy has the potential to offer protection against outcomes of air pollution exposure.

## 1. Introduction

In 2019, 99% of the global population lived in areas where the World Health Organization (WHO) air quality standards were not reached [1]. The WHO reported that air pollution accounted for around 7 million premature deaths per year owing to ischemic heart disease, stroke, chronic obstructive pulmonary disease, and lung cancer, as well as acute lower respiratory tract infections. Of those, approximately 89% were in low- and middle-income countries, with the highest prevalence in the WHO Southeast Asia and Western Pacific Regions [1].

The most frequently measured primary types of air pollution are particulate matter (PM) and gases (carbon monoxide (CO), nitrogen dioxide ($NO_2$), and sulfur dioxide ($SO_2$)). Secondary pollutants such as ozone ($O_3$) are produced through photochemical interaction between the primary pollutants and sunlight [2, 3].

PM is a complex combination that remains suspended in the atmosphere, irrespective of the particle's size. PM is typically classified based on its size, with two prevalent size fractions being PM10, also called coarse PM (from 2.5 μm to 10 μm), and PM2.5, also named fine PM ($< 2.5$ μm). Dust from the wind, volcanic eruptions, ocean spray, forest fires, and bioaerosols are the main non-combustion sources of PM [4]. Crushing and gritting processes, as well as dust that is released by vehicles and roads, are sources of PM10. Contrarily, PM2.5 is a byproduct of all forms of combustion, including those in cars, domestic wood burning, agricultural burning, forest fires, and some industrial activities. Particle weight and composition, as well as host parameters regulating the location and density of deposition in the respiratory tract, all affect its toxicity [2, 5, 6]. $O_3$ is a highly reactive gas that is commonly a significant component of photochemical smog [7]. Many epidemiologic studies have reported the probable hazardous effects of air pollution on cardiopulmonary morbidity and mortality [8, 9].

Inhaling air pollution causes pulmonary oxidative stress, which causes inflammation, through a complex set of molecular events. These events include the generation of major reactive oxygen species (ROS) such as superoxide, hydroxyl radical, nitric oxide, and peroxynitrite [10, 11]. Research indicates that PM2.5 contributes to an oxidant-antioxidant imbalance [12]. Furthermore, acute $O_3$ exposure alters the structure of the lung, disrupting the alveolar, epithelial barrier and causing type II alveolar epithelial cells to enlarge and hyperplasia. The migration of inflammatory cells into the lung after exposure to $O_3$ can also cause tissue damage due to the production of toxic mediators from activated macrophages and neutrophils, such as cytokines, ROS, nitrogen species, and proteolytic enzymes [13–15]. Therefore, several clinical indicators, including the spirometric indices, induced sputum, and bronchoalveolar lavage have been used to address the detrimental effects of air pollutants on the structure and function of the airways and lungs [16, 17].

In addition, there is mounting evidence from both clinical and epidemiological studies linking air pollution to cardiovascular disease. Several negative health effects, including hypertension, heart disease, stroke, and high blood pressure, are closely linked with PM2.5 and PM10 air pollution levels [18]. A study found that for every 10.5 μg/m3 of PM2.5, the risk of ischemic

heart disease, heart failure, arrhythmias, and cardiac arrest increases by 8~18%. This is likely due to mechanisms such as oxidative stress, systemic inflammation, accelerated atherosclerosis, and affected cardiac autonomic function [19, 20]. Therefore, changes in outcomes like blood pressure, heart rate (HR), and heart rate variability (HRV) which provide insight into cardiac hemodynamic and autonomic nervous system functioning are often assessed as cardiovascular markers of air pollution-induced toxicity [21, 22]. Additionally, measures of brachial artery diameter (BAD) and flow-mediated dilation (FMD), as the indicators of endothelial dysfunction, as well as blood lipid profiles which are associated with the development of atherosclerosis, are being extensively used in pollution-related cardiovascular research [23, 24].

The role of diet in mitigating the toxicity of air pollution has been gaining attention. Several studies have shown that taking dietary antioxidant supplements, such as fish oil, vitamin E, vitamin C, and vitamin B gives cardiopulmonary protection against air pollution [25–28]. The benefactor properties of these agents have been attributed mainly to their antioxidant and anti-inflammatory properties [3, 29], which can potentially combat the pro-inflammatory effects caused by pollutants. This systematic review aims to assess the role of dietary supplementation in reducing the cardiopulmonary impacts of air pollution toxicity.

## 2. Methods

This study was performed and reported following the Preferred Reporting Items for Systematic Reviews and Meta-Analyses (PRISMA) statement (S1 Checklist) [30]. The protocol for this study was prospectively registered in the International Prospective Register of Systematic Reviews (PROSPERO; registration ID: *CRD42023440510*).

### 2.1. Search strategy

A systematic search was conducted in PubMed/Medline, EMBASE, the Cochrane Central Register of Controlled Trials (CENTRAL), and ClinicalTrials.gov for clinical trials reporting the effects of dietary supplementations against cardiopulmonary effects of air pollution from January 1, 2000, up to March 31, 2023. Lists of references in selected articles were hand-searched to identify further studies. Search keywords and queries are available in the S1 Table.

### 2.2. Study selection

The records found through databases were merged, and the duplicate records were removed using EndNote X8 (Thomson Reuters, Toronto, ON, Canada). Studies were included if they matched the following criteria: (a) clinical trials published in peer-reviewed journals, (b) the intervention included dietary supplementations [including vitamin C, vitamin E (or its isoform γ-tocopherol), vitamin B, L-arginine, fish oil (n-3 polyunsaturated fatty acid (PUFA)), olive oil, sulforaphane, and n-acetylcysteine (NAC)], (c) the outcomes were cardiopulmonary assessments [including systolic blood pressure (SBP) and diastolic blood pressure (DBP), HR, blood lipid profile, BAD and FMD of brachial artery, HRV, Spirometric indices and nasal lavage, bronchoalveolar lavage or induced sputum markers (macrophage, polymorphonuclear neutrophil (PMN), and eosinophil)], (d) the study population were individuals exposed to air pollutants [including particulate matter (PM), diesel exhaust particles (DEPs), $O_3$, NO2, and SO2]. Review articles, systematic reviews, meta-analyses, and studies with non-trial design (including cross-sectional, cohort, and case-control studies), in addition to non-English studies, conference abstracts, and posters were excluded. Articles were screened in two stages. In the first stage, two reviewers (MI and MS) independently reviewed titles and abstracts and selected the studies that matched the criteria for full-text evaluation. Discrepancies were

discussed with a third reviewer (MZ). In the second stage of screening through full-text appraisal, all clinical trials meeting the inclusion criteria were screened by the same authors. Disagreements were discussed and resolved between the reviewers until a consensus was reached.

### 2.3. Data extraction

Data was extracted by two independent reviewers (MI and MS) using a data extraction sheet. The following data was extracted: name of the first author, title of study, year of publication, region of study, study design, starting and ending date of trial, sample size, age of participants, gender distribution of participants, exposure (pollutant) type, exposure frequency, exposure dose, intervention type, intervention frequency, intervention dose, follow-up time, case definition, type of assessed outcome, mean difference and standard deviation (SD) of assessed outcome from the baseline to endpoint. In case of an inconsistency between the two reviewers, a third reviewer (MZ) intervened until a consensus was reached.

### 2.4. Risk of bias assessment

Two blinded reviewers (MI and MS) assessed the risk of bias using the revised Cochrane risk-of-bias tool for randomized trials (RoB 2) [31]. If there were any discrepancies, another reviewer (MZ) was consulted. RoB 2 provides separate tools for parallel trials and crossover trials. The RoB 2 tool for parallel trials assesses five domains, including the bias arising from the randomization process (domain 1), bias due to deviations from intended intervention (domain 2), bias due to missing outcome data (domain 3), bias in the measurement of the outcome (domain 4), and bias in the selection of the reported result (domain 5). The RoB 2 tool for crossover trials assesses the above-mentioned domains in addition to the bias arising from period and carryover effects (domain s). The *Robvis* web application [32] was utilized to visualize the results of the risk of bias assessment.

## 3. Results

The screening process of studies is illustrated in Fig 1. A total of 4681 non-duplicate records were screened for eligibility. After title/abstract screening, 41 studies were assessed for full-text evaluation, and 18 studies eventually met the inclusion criteria of the systematic review (Fig 1).

### 3.1. Study characteristics

Table 1 demonstrates the characteristics of all included studies. Studies had been conducted in the USA (n = 10), China (n = 3), Canada (n = 2), Mexico (n = 2), and Sweden (n = 1). Eight studies followed a parallel trial design, while ten were crossover trials (Table 1).

### 3.2. Risk of bias assessment in included studies

The risk of bias assessment for parallel and crossover trials is presented in Fig 2 and Table 2, respectively (Fig 2 and Table 2). Four studies [26–28, 33] were judged to have an overall high risk of bias. Among these studies, two studies [27, 33] had a high risk of bias attributed to domain 1 (randomization process), and two studies [26, 28] had a high risk of bias in domain 5 (selection of the reported result). Three studies [34–36] were judged to have some concerns in their overall risk of bias assessment. The other 11 studies [23, 25, 37–45] were judged to have a low risk of bias.

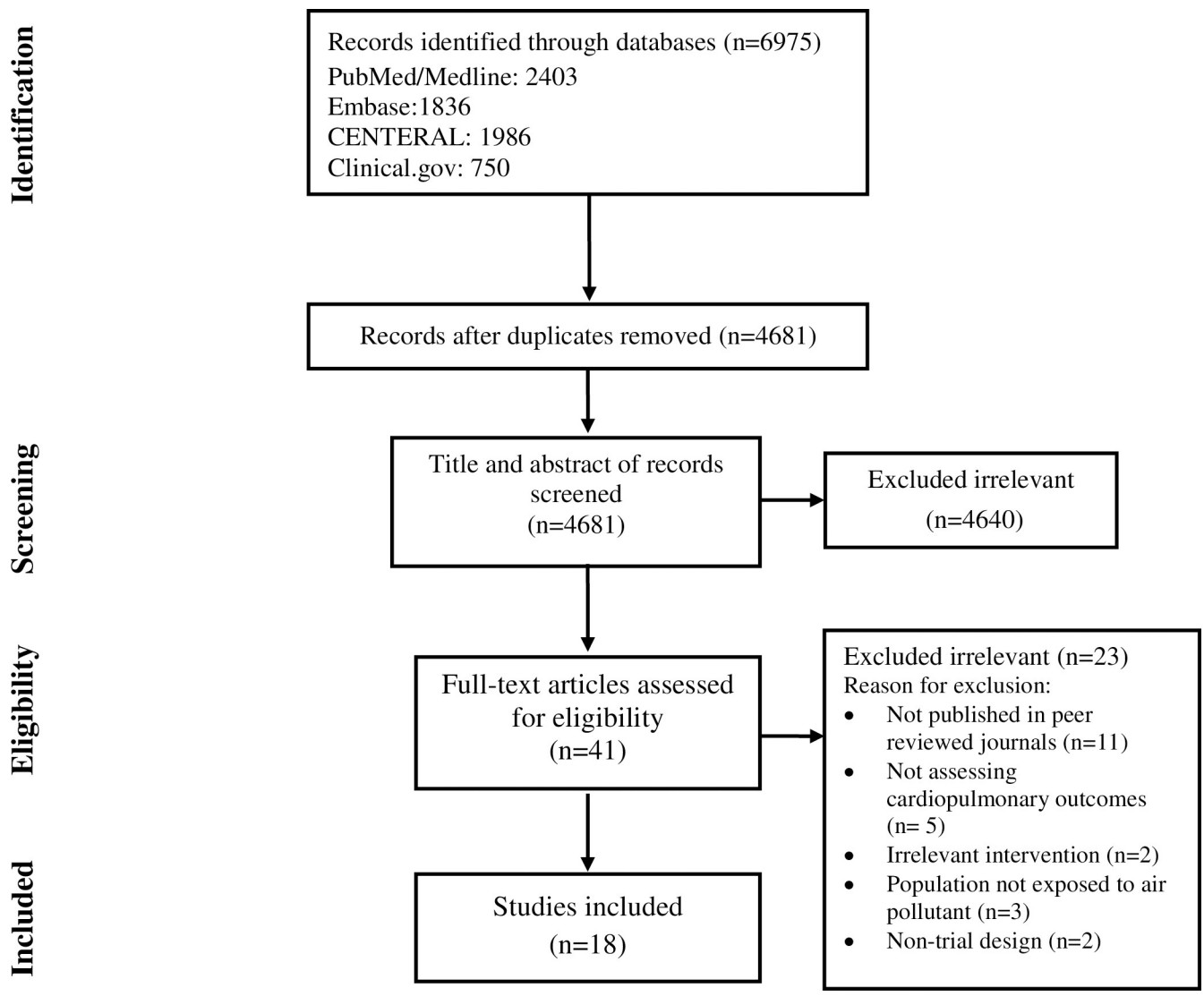

**Fig 1. Flow chart of study selection for inclusion in the systematic review.**

## 3.3. Characteristics of participants, pollutant exposure, dietary intervention, and assessed outcomes

The characteristics of the participants in the included studies are presented in Table 3. Three studies were conducted on asthmatic individuals [26, 28, 35], one study was conducted on hypertensive participants [38], and one study was performed on aged nursing home residents [40]. Other 13 studies were conducted on generally healthy individuals (Table 3).

Table 4 provides detailed data on the type of pollutant exposure, dietary interventions, and assessed outcomes across the included studies. Generally, ten studies evaluated the effects of dietary supplementations on cardiovascular outcomes (including BP, HR, HRV, lipid profile, FMD, or BAD) of air pollution exposure [23, 25, 27, 34, 37–40, 42, 43]. In 9 studies, the effects of dietary supplementations on pulmonary outcomes of air pollution toxicity (including Spirometric indices and airway inflammation) were assessed [26, 28, 33, 35–37, 41, 44, 45] (Table 4).

**Table 1. Characteristics of included studies.**

| Authors (Reference) | Publication year | Country | Trial design |
|---|---|---|---|
| Chen et al. [37] | 2022 | USA | Parallel |
| Ren et al. [42] | 2022 | China | Crossover |
| Li et al. [38] | 2021 | China | Parallel |
| Burbank et al. [35] | 2020 | USA | Crossover |
| Lin et al. [39] | 2019 | China | Parallel |
| Zhong et al. [27] | 2017 | Canada | Crossover |
| Sack et al. [43] | 2016 | USA | Crossover |
| Duran et al. [36] | 2016 | USA | Crossover |
| Tong et al. [25] | 2015 | USA | Parallel |
| Carlsten et al. [44] | 2014 | Canada | Crossover |
| Heber et al. [33] | 2014 | USA | Crossover |
| Tong et al. [34] | 2012 | USA | Parallel |
| Brook et al. [23] | 2009 | USA | Crossover |
| Mudway et al. [45] | 2006 | Sweden | Crossover |
| Romieu et al. [40] | 2005 | Mexico | Parallel |
| Romieu et al. [28] | 2002 | Mexico | Parallel |
| Trenca et al. [26] | 2001 | USA | Crossover |
| Samet et al. [41] | 2001 | USA | Parallel |

The dietary interventions were fish oil in 5 studies [25, 34, 37, 39, 40], olive oil in 3 studies [25, 34, 37], vitamin C in 2 studies [23, 42], co-supplementation of vitamin C and vitamin E (α-tocopherol) in 4 studies [26, 28, 41, 45], γ-tocopherol (vitamin E isoform) in 1 study [35], vitamin B (B6, B12, and folic acid) in 1 study [27], NAC in 1 study [44], co-supplementation of vitamin C and NAC in 1 study [43], L-arginine in 1 study [38], and sulforaphane in 2 studies [33, 36] (Table 4).

## 3.4. Effects of dietary supplementations on cardiovascular outcomes of air pollution exposure

**3.4.1. Blood pressure.** Overall, BP was the assessed outcome in 8 studies. Three studies assessed the impact of fish oil supplementation on BP among individuals in contact with air pollutants [25, 37, 39]. In one study, SBP and DBP markedly elevated 20-hr post-pollutant exposure ($O_3$, $300 \pm 30$ ppb) in the unsupplemented group (p<0.05), while in participants receiving fish oil (3 g/day, four weeks), no increase in SBP and DBP was observed [37]. In another study, pollutant exposure (PM2.5, 38 μg/m$^3$) was not associated with SBP and DBP in placebo and fish oil (2.5 g/day, four months) groups and there were no remarkable differences between the groups [39]. The findings of the other study also showed that both the unsupplemented group and the fish oil group (3 g/day, 28 days) had a non-significant increase in SBP and a significant increase in DBP after PM2.5 exposure [25].

Two studies evaluated the effects of olive oil supplementation on BP in participants exposed to air pollution. One study showed that the SBP and DBP significantly increased 20-hr after pollutant ($O_3$, $300 \pm 30$ ppb) exposure in the unsupplemented group (p<0.05), while in participants receiving olive oil (3 g/day, four weeks), a significant decrease was reported in SBP (p<0.05) after pollutant exposure, while also DBP did not increase after the exposure [37]. However, the findings of the other study demonstrated that both the unsupplemented group and the olive oil group (3 g/day, 28 days) group had a non-significant increase in SBP and a significant increase in DBP post-PM2.5 exposure [25].

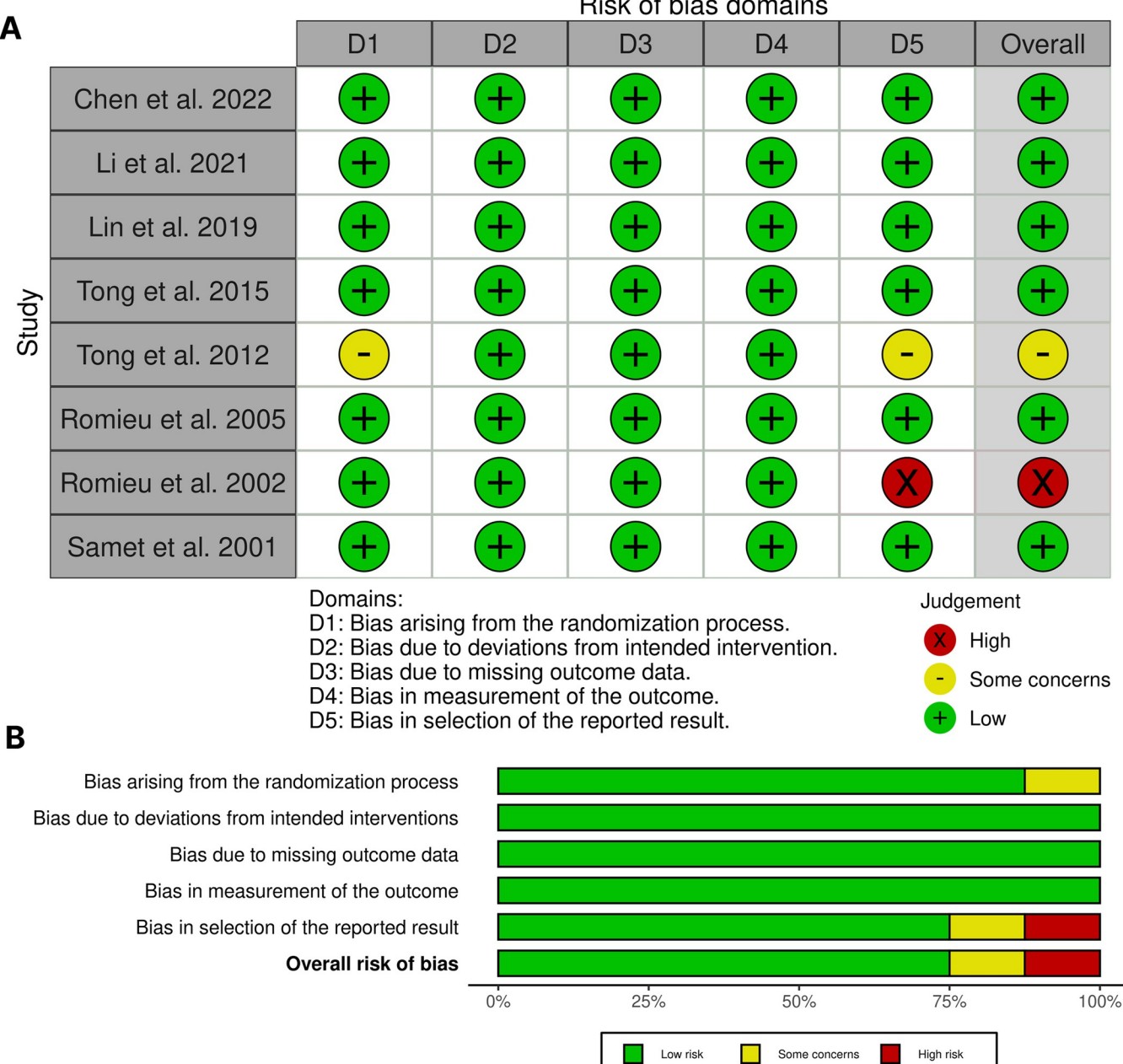

**Fig 2. Summary of risk of bias assessments of included parallel trials.** **(A)** Risk of bias in each included study according to RoB 2 **(B)** Summary of judgments about each domain of RoB 2 is presented as percentages in all included parallel trials.

Two studies reported the effects of supplementation with vitamin C on the BP of air pollutant-exposed participants [23, 42]. One study showed that vitamin C supplementation (2000 mg/day, one week) was linked to a 3.37% [95% CI: - 5.22%, - 1.52%] decline in SBP in comparison with the placebo group, while no significant difference was seen in DBP [42]. Nevertheless, in the other study, vitamin C supplementation (2000 mg single dose before exposure) did not blunt the pollutant-induced changes in BP compared to the placebo group [23].

One study assessed the impact of L-arginine supplementation on BP of individuals exposed to air pollutants, showing that in contrast to the placebo group, individuals receiving

**Table 2. Risk of bias assessment for cross-over trials according to RoB 2 tool.**

| Study | D1 | Ds | D2 | D3 | D4 | D5 | Overall |
|---|---|---|---|---|---|---|---|
| Ren et al. 2022 | Low | Low | Low | Low | Low | Low | Low |
| Burbank et al. 2020 | Some Concerns | Low | Some Concerns | Low | low | Low | Some Concerns |
| Zhong et al. 2017 | High | Low | Low | Low | Low | Low | High |
| Sack et al. 2016 | Low | Low | Low | Low | Low | Low | Low |
| Duran et al. 2016 | Some Concerns | Low | Some Concerns | Low | Low | Low | Some Concerns |
| Carlsten et al. 2014 | Low | Low | Low | Low | Low | Low | Low |
| Heber et al. 2014 | High | Low | Some Concerns | Low | Low | Low | High |
| Brook et al. 2009 | Low | Low | Low | Low | Low | Low | Low |
| Mudway et al. 2006 | Low | Low | Low | Low | Low | Low | Low |
| Trenca et al. 2001 | Low | Low | Low | Low | Low | High | High |

**D1:** Bias arising from the randomization process; **D2:** Bias due to deviations from intended intervention; **Ds:** Bias arising from period and carryover effects; **D3:** Bias due to missing outcome data; **D4:** Bias in measurement of the outcome; **D5:** Bias in the selection of the reported results.

L-arginine (four 0.75 g pills three times a day for two weeks) had a remarkable decline of 5.3 mmHg [95% CI: - 9.9, - 0.7] and 4.3 mmHg [95% CI: - 7.2, - 1.3] mmHg in the alterations of resting SBP and DBP at 30 min post-exposure, respectively [38].

**3.4.2. Heart rate.** HR was the assessed outcome in 3 studies. One study assessed the effects of vitamin B (B6, B12, and folic acid) supplementation on HR among participants exposed to air pollutants. Findings demonstrated that without vitamin B supplementation (one tablet daily containing 2.5 mg folic acid, 50 mg vitamin B6, and 1 mg vitamin B12 for four weeks), HR increased after PM2.5 exposure. Subsequent to B vitamin supplementation, the association of PM2.5 with post-exposure HR was significantly attenuated ($p = 0.003$). The reduction of the PM2.5-induced HR by vitamin B was still significant at 24 h post-exposure [27].

One study assessed the impact of fish oil (3g/day, four weeks) or olive oil (3g/day, four weeks) supplementation on changes in HR after pollutant ($O_3$, 300 ± 30 ppb) exposure but did not find any modifying effects of these interventions on HR [37].

Another study also assessed the impact of vitamin C supplementation (2000 mg single dose before exposure) on the HR of air pollutant-exposed ($O_3$ and PM2.5) individuals, showing that HR increased to a statistically similar degree in vitamin C and placebo groups [23].

**3.4.3. Heart rate variability.** Overall, HRV was assessed in 5 studies. Three studies assessed the impact of fish oil supplementation on HRV among individuals exposed to air pollutants [34, 37, 40]. One study did not report any significant changes in HRV parameters that could be attributed to the fish oil (3g/day, four weeks) intervention [37]. However, another study showed that supplementation with fish oil (3g/day, four weeks) ameliorated the pollutant-induced reductions in high-frequency/low-frequency (HF/LF) ratio, as well as heightened normalized low-frequency (nLF) HRV [34]. Moreover, another study showed that fish oil (2g/day, five months) supplementation prevented HRV reduction associated with PM2.5 exposure so that in individuals taking fish oil, the decline in HRV–high-frequency $\log_{10}$-transformed related to a 1-SD alteration in PM2.5 was −54% [95% CI: −72, −24] before supplementation and only −7% [95% CI: −20,+7] within the supplementation phase ($p < 0.01$ for supplementation's effect), with alterations in other HRV indices also being remarkably less noticeable while taking supplements [40].

In one study, the effects of vitamin B (one vitamin B tablet daily containing 2.5 mg folic acid, 50 mg vitamin B6, and 1 mg vitamin B12 for four weeks) supplementation on HRV were evaluated [27]. Findings demonstrated that without vitamin B supplement, HRV parameters,

**Table 3. Characteristics of participants across included studies.**

| Author | Age* | Participants | Total study population (Female/Male) | Intervention group population (Female/Male) | Control population (Female/Male) |
|---|---|---|---|---|---|
| Chen et al. [37] | Control: 24.2±4.5, Intervention 1: 26.6±3.9, Intervention 2: 26.8±3.9 | Healthy | 43 (20/23) | Intervention 1: 15 (8/7), Intervention 2: 16 (6/10) | 12 (6/6) |
| Ren et al. [42] | Total participants: 20.1±3.0 | Healthy | 58 (24/34) | 28 | 30 |
| Li et al. [38] | Control: 63.8±5.7, Intervention: 63.5±5.2 | Current non-smoker adults with elevated blood pressure | 98 (40/58) | 49 (20/29) | 49 (20/29) |
| Burbank et al. [35] | Total participants: 23 (median) | Adults with mild intermittent allergic asthma | 15 (11/4) | - | - |
| Lin et al. [39] | Control: 22.87±1.28 / Intervention: 23.03 ±2.26 | Healthy | 65 (38/27) | 34 (20/14) | 31 (18/13) |
| Zhong et al. [27] | Total participants: 18–60 (range) | Healthy | 10 (6/4) | - | - |
| Sack et al. [43] | Total participants: 28.6 | Healthy | 21 (8/13) | - | - |
| Duran et al. [36] | Total participants: 18–50 (range) | Healthy | 15 | - | - |
| Tong et al. [25] | Total participants: 58±1, Intervention 1 (Olive oil): 59.3±1.1, Intervention 2 (Fish oil): 57.4±1.4, Naïve: 57.8±1.3 | Healthy | 42 (32/10) | Intervention 1 (Olive oil): 13 (9/4), Intervention 2 (Fish oil): 16 (12/4) | 13 (11/2) |
| Carlsten et al. [44] | Total: 29±8 | Generally healthy (with or without baseline airway responsiveness) | 26 (13/13) | - | - |
| Heber et al. [33] | Above 18 (range) | Healthy subjects positive for cat allergens | 28 (14/14) | - | - |
| Tong et al. [34] | Total participants: 58±1, Intervention: 57.4 ±1.4, Control:59.3±1.1 | Healthy | 29 (21/8) | 16 (12/4) | 13(9/4) |
| Brook et al. [23] | Total participants: 27±8 | Healthy | 50 (31/19) | - | - |
| Mudway et al. [45] | Total participants: 24.1±2.6 | Healthy and ozone-sensitive | 14 | - | - |
| Romieu et al. [40] | Total participants: 81.96, Intervention:81, Control:83 | Nursing home residents | 50 (34/16) | 26 (17/9) | 24 (17/7) |
| Romieu et al. [28] | Total participants:8.74, Intervention:8.6, Control:8.9 | Asthmatic children | 158 (56/102) | 80 (27/53) | 78 (29/49) |
| Trenca et al. [26] | Total participants: 27±6.29 | Adult subjects sensitive to SO2 with asthma | 17 (12/5) | - | - |
| Samet et al. [41] | Total participants: 26.74, Intervention: 26.9 ±3.2, Control: 26.6 ±4.0 | Healthy | 31 (3/28) | 15 (2/13) | 16 (1/15) |

* Age is expressed as mean (±SD), unless otherwise specified.

including standard deviation of normal-to-normal intervals (SDNN), LF/HF ratio, and LF power decreased after pollutant exposure. After vitamin B supplementation, the associations of PM2.5 with HRV ($P_{intervention}$ = 0.01 for LF) were significantly decreased. Moreover, supplementation with vitamin B decreased the effect size of pollutants by 96% for LF/HF ratio and 90% for LF. Additionally, even though it did not reach statistical significance, vitamin B supplementation decreased the pollutant impact on SDNN by 57% [27].

Two studies investigated the effects of olive oil supplementation (3g/day, four weeks) on HRV among pollutant-exposed participants [34, 37]. Neither of the studies found any significant modifying effects of olive oil supplementation on HRV parameters.

**Table 4. Characteristics of pollutant exposure, intervention and assessed outcomes in included studies.**

| Authors (Reference) | Pollutant exposure type | Pollutant exposure dose | Pollutant exposure frequency | Dietary intervention | Intervention dose | Intervention duration | Assessed Outcome (s) |
|---|---|---|---|---|---|---|---|
| Chen et al. [37] | O3 | 300 ± 30 ppb | Two hours single exposure | Fish Oil or Olive Oil | Fish Oil: 3g/day, Olive Oil: 3g/day | Four weeks | Spirometric indices, induced sputum, BP, HR, HRV, BAD, FMD, lipids |
| Ren et al. [42] | PM2.5, PM10 | PM2.5: 164.91 µg/m3, PM10: 327.05 µg/m3 | Ambient free air | Vitamin C | Four pills (2000 mg) of vitamin C daily | One week with a 2-week washout period | BP, lipids |
| Li et al. [38] | PM2.5, Black carbon, NO2 | Intervention group: PM2.5: 63±62 µg/m3, Black carbon: 2.9 ±2.6 µg/m3, NO2: 87.9±36.8 µg/m3 Placebo group: PM2.5: 65.9±64.1 µg/m3, Black carbon: 2.9 ±2.3 µg/m3, NO2: 88.2±34.3 µg/m3 | Ambient free air | L-arginine | Four pills of L-arginine (each contained 0.75 g L-arginine) 3 times a day | Two weeks | BP |
| Burbank et al. [35] | O3 | 0.25 ppm | Three hours single exposure | Gamma tocopherol (vitamin E isoform) | Two tabs (each containing 600 mg gamma tocopherol) every 12 hours | Four doses, with the final dose administered the morning of O3 exposure | Induced sputum |
| Lin et al. [39] | PM2.5 | 38 µg/m3 | Ambient free air | Fish Oil | Two fish oil capsules (1.25 g each) every day | Four months | BP |
| Zhong et al. [27] | PM2.5 | 250 µg/m3 | Two hours single exposure | Vitamin B (folic acid, B6, B12) | One vitamin B tablet daily (2.5 mg folic acid, 50 mg vitamin B6, and 1 mg vitamin B12) | Four weeks | HR, HRV |
| Sack et al. [43] | PM2.5 | 200 µg /m3 | Two hours single exposure | Vitamin C + NAC | 500-mg vitamin C twice daily for 7 days and two 600-mg NAC capsules taken twice on the day before the exposure session. On the morning of the exposure session, subjects were administered 1000 mg vitamin C and 600 mg NAC. | One week | BAD, FMD |
| Duran et al. [36] | O3 | 0.4 ppm | Two hours single exposure | Sulforaphane | 200 g of broccoli sprout homogenate (BSH) daily | Once daily for 3 days during the initial study period, and the alternate treatment during the crossover period. | Induced sputum |
| Tong et al. [25] | PM2.5 | 253 ± 16 µg/m3 | Two hours single exposure | Fish Oil or Olive Oil | Olive Oil: 3 g/day Fish Oil: 3 g/day | Daily for 28 days | BP, FMD, BAD, lipids |
| Carlsten et al. [44] | PM2.5 | 300 µg/m3 | Two hours single exposure | NAC | NAC capsule 600 mg three times a day | Six days with minimum 2 -week washout | Spirometric indices, induced sputum |
| Heber et al. [33] | DEP (PM1) | 300 µg in 200 µL saline | DEP challenge equivalent to 40h exposure to ambient polluted air | Sulforaphane | 100 µmol sulforaphane daily | Four days | Nasal lavage (nasal WBC count) |
| Tong et al. [34] | PM2.5, PM0.1 | 278 ± 19 µg/m3 | Two hours single exposure | Fish Oil or Olive Oil | Olive Oil: 3 g/day Fish Oil: 3 g/day | Four weeks | HRV, lipids |

(*Continued*)

**Table 4.** (Continued)

| Authors (Reference) | Pollutant exposure type | Pollutant exposure dose | Pollutant exposure frequency | Dietary intervention | Intervention dose | Intervention duration | Assessed Outcome (s) |
|---|---|---|---|---|---|---|---|
| Brook et al. [23] | O3, PM2.5 | O3:120ppb, PM2.5:150 µg/m3 | Two hours single exposure | Vitamin C | 2000 mg | Single dose (2 hours before exposure) | BP, HR, BAD, FMD |
| Mudway et al. [45] | O3 | 0.2 ppm | Two hours single exposure | Vitamin C + Vitamin E (α-tocopherol) | Vitamin C: 500 mg/day, Vitamin E: 100 mg/day | One week | Spirometric indices, BAL |
| Romieu et al. [40] | PM2.5 | 18.6±7.95 µg/m3 | Ambient free air | Fish Oil | 2 g/day | Five months | HRV |
| Romieu et al. [28] | O3, NO2, PM10 | O3: 102±47ppb, NO2: 30±15ppb, PM10: 56.68±27.36 µg/m3 | Ambient free air | Vitamin C + Vitamin E (α-tocopherol) | Vitamin C: 250 mg/day, Vitamin E: 50 mg/day | Twelve weeks | Spirometric indices |
| Trenca et al. [26] | O3, SO2 | O3: 0.12ppm, SO2:0.1ppm and 0.25ppm | O3 (45min), SO2 (two exposures each 10min) | Vitamin C + Vitamin E (α-tocopherol) | Vitamin C: 500 mg/day, Vitamin E: 400 IU/day | Five weeks | Spirometric indices |
| Samet et al. [41] | O3 | 0.4 ppm | Two hours single exposure | Vitamin C + Vitamin E (α-tocopherol) + Vegetable (carotenoid) | Vitamin C: 250 mg/day, Vitamin E: 50 IU/day, Vegetable: 12 oz | Two weeks | Spirometric indices, BAL |

**O3:** Ozone; **PM:** Particulate matter; **ppb:** parts per billion, **ppm:** parts per million; **NAC:** N-acetyl cysteine; **BP:** Blood pressure, **BAD:** Brachial artery diameter, **FMD:** Flow-mediated dilation, **BAL:** Bronchoalveolar lavage; **HR:** Heart rate; **HRV:** Heart rate variability.

**3.4.4. Brachial artery diameter and flow-mediated dilation.** In total, BAD or FMD was the assessed outcome in 4 studies. Two studies evaluated the outcomes of fish oil supplementation on BAD or FMD of individuals exposed to air pollution [25, 37]. One study did not find acute effects of pollutant (O3, 300 ± 30 ppb) on BAD and FMD, nor found remarkable differences between fish oil (3 g/day, four weeks) supplements and the control group [37]. Similarly, the other study also showed that fish oil (3 g/day, 28 days) did not affect the BAD and FMD changes attributed to PM2.5 [25].

Two studies assessed the effects of olive oil supplementation on air pollution-induced changes in BAD or FMD [25, 37]. One study found no effects of pollutant (O3, 300 ± 30 ppb) on BAD and FMD and no noticeable disparities between the olive oil (3 g/day, four weeks) and control groups [37]. However, the other study demonstrated that olive oil (3 g/day, 28 days) blunted the FMD changes attributed to PM2.5 [25]. In other words, the study showed that FMD was significantly lower after pollutant exposure in the unsupplemented group (–19.4% average decrease compared to baseline per 100-µg/m3 increase in pollutant concentration; 95% CI: –36.4%, –2.3%; p = 0.03), while in the olive oil group, exposure to pollutant led to a smaller non-significant decrease in FMD (–7.6%; 95% CI: –21.5%, 6.3%; p = 0.27) [25].

One study evaluated the impact of vitamin C (2000 mg, single dose) on BAD and FMD of air pollution-exposed (O3 and PM2.5) individuals but did not find changes after exposure or differences among the supplement and control groups [23].

In one study, the effects of the combination of vitamin C (500 mg twice daily for 7 days before exposure and 1000 mg on the morning of exposure) and NAC (two 600-mg capsules twice the day before exposure and 600 mg on the morning of exposure) on BAD and FMD were assessed [43]. Neither vitamin C + NAC pretreatment nor PM2.5 exposure resulted in a statistically significant alteration in FMD. However, pollutant exposure was linked to vasoconstriction with a decline in BAD of 0.09 mm [95% CI—0.01–0.17 mm; P = 0.03). Moreover,

pretreatment with vitamin C + NAC modified the effect of pollutants on BAD. Interestingly, exposure to pollutants exhibited higher vasoconstriction in individuals pretreated with vitamin C + NAC, in contrast to participants who took the placebo, with a pollutant effect of 20.18 mm [95% CI, 20.28 to 20.07 mm; P = 0.01] in comparison with 20.01 [95% CI, 20.11 to 0.9 mm; P = 0.81], respectively [43].

**3.4.5. Lipid profile.** A total of 4 studies assessed blood lipids as the trial outcome. Three studies addressed the impact of fish oil on the lipid markers of subjects exposed to air pollution [25, 34, 37]. One study showed that the average post-exposure triglyceride (TG) and very low-density lipoprotein (VLDL) concentrations were lower in the fish oil (3 g/day, four weeks) group in comparison with the control group (p<0.01). Nevertheless, no difference in other blood lipid levels, including high-density lipoprotein (HDL), low-density lipoprotein (LDL), and total cholesterol (TC) was reported [37]. Another study showed that pollutant (PM2.5 and PM0.1) exposure did not cause a sudden rise in TG and VLDL in the fish oil-supplemented (3 g/day, four weeks) group, while these lipids increased significantly in the control group (olive oil) after the pollutant exposure, suggesting that fish oil appears to attenuate pollutant-induced increases in TG and VLDL [34]. However, another study also failed to show any significant effects of PM2.5 exposure and fish oil (3 g/day, 28 days) supplementation on TG, TC, HDL, LDL, and VLDL [25].

Three studies evaluated the impact of olive oil supplementation on the blood lipids of individuals exposed to air pollution [25, 34, 37]. Olive oil (3 g/day, four weeks) supplementation was not linked to changes in TG, TC, LDL, HDL, and VLDL compared to the control group in one study [37]. Similarly, another study demonstrated no significant effects of PM2.5 exposure and olive oil supplementation (3 g/day, 28 days) on TG, TC, HDL, LDL, and VLDL [25]. Furthermore, another study showed that olive oil (3 g/day, four weeks) supplementation failed to blunt the pollutant-induced increase in TG and VLDL, while these increments were blunted by fish oil (3 g/day, four weeks) [34].

One study assessed the effects of vitamin C (2000 mg daily for one week) on the blood lipid levels of air pollution-exposed (PM2.5 and PM10) subjects [42]. No remarkable difference between vitamin C and placebo groups was found in terms of TG, TC, HDL, LDL, apolipoprotein A (APOA), and apolipoprotein B (APOB). Nevertheless, in contrast to the placebo group, vitamin C intake was accompanied by a 6.28% [95% CI:0.29%, 12.27%] rise in APOB levels in female participants only [42].

## 3.5. Effects of dietary supplementations on pulmonary outcomes of air pollution exposure

**3.5.1. Spirometric indices.** Overall, Spirometric findings were assessed in six studies. Four studies assessed the impact of vitamin C and vitamin E co-supplementation on Spirometric indices among pollutant-exposed individuals [26, 28, 41, 45]. One study demonstrated that exposure to $O_3$ (0.2 ppm) led to a remarkable decline in the forced expiratory volume (FEV1) with no protective effects occurring after vitamin (vitamin C: 500 mg/day and vitamin E: 100 mg/day for one week) supplementation (-8.5%) compared to placebo (-7.3%) treatment [45]. However, another study showed that in children diagnosed with moderate and severe asthma, exposure to $O_3$ was negatively correlated with the forced expiratory flow (FEF25–75) (-13.32 ml/ second/10 ppb; p < 0.001), FEV1 (-4.59 ml/10 ppb; p = 0.036), and peak expiratory flow (PEF) (-15.01 ml/second/10 ppb; p = 0.04) in the group receiving placebo, while vitamin C and vitamin E co-supplementation (vitamin C: 250 mg/day and vitamin E: 50 mg/day for twelve weeks) blunted these changes [28]. Similarly, another study showed that asthmatic individuals who took dietary vitamin C and vitamin E (vitamin C: 500 mg/day and vitamin E: 400 IU/day

for five weeks) had a less severe response to sulfur dioxide than those given a placebo (i.e., FEV1: -1 2% vs. 4.4%; PEF: +2.2% vs.-3.0%; mid-forced expiratory flow: +2.0% vs. 4.3%, respectively) [26]. Furthermore, findings from another study also revealed that $O_3$ (0.4 ppm) exposure resulted in reductions in forced vital capacity (FVC) and FEV1 values that were 24% (p = 0.046) and 30% (p = 0.055) lower in the group supplemented with a combination of vitamin C, vitamin E, and vegetable (vitamin C: 250 mg/day, vitamin E: 50 IU/day, and vegetable: 12 oz for two weeks) than in the placebo group [41].

One study assessed the effects of fish oil (3 mg/day, four weeks) or olive oil (3 mg/day, four weeks) supplementation on Spirometric findings of individuals exposed to $O_3$ (300 ± 30 ppb) [37], showing that the FEV1/FVC (p = 0.0004) and normalized FEV1 (p = 0.005) in the fish oil group were remarkably elevated compared with those in the control group post- $O_3$ exposure. The $O_3$-induced decrease in the FEV1/FVC ratio observed in the control group was remarkably prevented by 70% in the fish oil group (p = 0.01). Moreover, the $O_3$-induced decrease in the FVC and FEV1 in the control group was non-significantly decreased by 19% (p = 0.89) and 48% (p = 0.11) by fish oil supplementation, respectively. On the other hand, olive oil supplementation also had a non-significant 34% protection against $O_3$-induced decrease in the FEV1/FVC ratio (p = 0.31) [37].

One study evaluated the impact of NAC supplementation (600 mg three times a day for six days) on Spirometric indices of PM2.5-exposed subjects, demonstrating that in hyper-responsive individuals, airway responsiveness (characterized by the dose-response slope of % fall in FEV1) increased by 42% following pollutant exposure compared with filtered air (p = 0.03) and NAC supplementation reversed this increase [44].

**3.5.2. Airway inflammation.** In total, airway inflammation was evaluated in seven studies. Two studies evaluated the effects of vitamin C and vitamin E co-supplementation on airway inflammation of pollutant-exposed participants [41, 45]. One of the studies showed that $O_3$ (0.2 ppm) exposure was associated with a noticeable influx of neutrophils into the airways after both placebo and vitamin (vitamin C: 500 mg/day and vitamin E: 100 mg/day for one week) groups without differences between the groups [45]. Similarly, the other study found that the levels of neutrophils in the bronchoalveolar lavage fluid post- $O_3$ (0.4 ppm) exposure were significantly increased for both the vitamin-supplemented (vitamin C: 250 mg/day, vitamin E: 50 IU/day, and vegetable: 12 oz for two weeks) and placebo groups, but the magnitudes of these increments did not differ between the two groups [41].

Two studies addressed the effects of sulforaphane on airway inflammation attributed to air pollution toxicity [33, 36]. One study showed that $O_3$ (0.4 ppm) exposure remarkably increased the number of neutrophils in sputum in placebo and sulforaphane (200 g broccoli sprout homogenate daily for 3 days) groups; however, the supplementation group had no noticeable difference in sputum neutrophilia in comparison with placebo [36]. However, in the other study, it was demonstrated that the nasal lavage WBC counts decreased by 54% when the diesel exhaust particle challenge was preceded by daily sulforaphane (100 μmol daily for four days) administration [33].

One study evaluated the effects of fish oil and olive oil supplementation on airway inflammation of participants exposed to air pollutants, showing that both fish oil (3 g/day, four weeks) or olive oil oil (3 g/day, four weeks) supplementation did not noticeably modify the rise in PMN% or decrease in macrophage observed in induced sputum after $O_3$ (300 ± 30 ppb) exposure [37].

One study assessed the effects of NAC (600 mg three times a day for six days) supplementation on airway inflammation of PM2.5-exposed subjects and showed that the sputum bronchial epithelial cells in hyper-responsive individuals increased after pollutant exposure

compared to filtered exposure and this effect was attenuated by NAC supplementation with a borderline significance [44].

In one study, the effects of γ-tocopherol (two 600-mg tablets every 12 hours for four doses) on airway inflammation of $O_3$-exposed (0.25 ppm) mild asthmatic individuals were explored [35]. Findings showed that a brief duration of γ-tocopherol did not ameliorate baseline eosinophilic airway inflammation nor $O_3$-induced neutrophilic inflammation in these individuals.

## 4. Discussion

Nutrition is well known for its importance in the prevention and treatment of chronic disorders [46, 47]. Our result generally showed the potential beneficial clinical effects of anti-inflammatory and antioxidant dietary supplements against air pollution toxicity. Besides dietary supplements, previous reviews have highlighted several pharmacological agents, including beta-blockers, statins, and endothelin inhibitors for their protective effects against air pollution [48]. Moreover, antioxidant-rich diets have also been reviewed in terms of their potential in mitigating the burden of traffic-related air pollution [29]. However, as far as we are concerned, this study is the first systematic review of how dietary supplements modify the impact of air pollution on the cardiovascular system and the lungs. This review incorporated data from clinical trials to propose a strategy for ameliorating the consequences of air pollution toxicity. We separately discuss the effects of vitamins, minerals, and botanical compounds on the adverse effects of the most frequent types of air pollution.

### 4.1. Fish oil and olive oil supplements

The findings of our systematic review generally showed that consumption of fish oil and olive oil could have a positive effect on spirometric indices and BP in healthy individuals exposed to $O_3$. Also, the efficacy of fish oil in modulating TG levels has been observed [37]. Besides, fish oil supplementation has beneficial effects on lipid profile alterations and HRV caused by PM2.5 inhalation, while olive oil can be relatively effective in decreasing the effects of air pollution on FMD [25, 34, 40].

Similarly, in line with clinical findings, some animal studies have shown that omega-3-containing oils reduce the inflammation caused by fine PM pollution, while fish oil protects against O3-induced vascular damage [49, 50]. The biological impacts of fish oil are thought to arise from their composition of n-3 PUFA, particularly docosahexaenoic acid (DHA) and eicosapentaenoic acid (EPA). Several studies have shown that the metabolization of n-3 PUFA can modulate inflammation by bioactive lipid mediators such as resolvins, protectins, and 5-series leukotrienes. Moreover, EPA and DHA have been shown to stimulate the antioxidant genes HMOX1 and GPX1 and turn down the activity of the pro-inflammatory factor NF-κB, which is a critical transcription factor in air pollution-induced inflammation. Studies have shown that in human atrial myocardium, peroxisome proliferator-activated receptor (PPAR) activation and increased expression of genes related to fatty acid metabolism, as well as oxidation of mitochondrial fatty acids, and antioxidant capacity occur after treatment with high-dose n-3 PUFA. Therefore, EPA and DHA may help mitigate the inflammation and oxidative stress brought on by air pollution exposure [51–54]. Taken together, the findings from studies in this review demonstrate that fish oil supplementation was able to attenuate the pollutant-induced alterations in lipid profile, suggesting the cardioprotective effects of n-3 PUFAs may be mediated through their ability to modulate lipid metabolism. Additionally, the review found that fish oil supplementation prevented the reductions in HRV parameters associated with air pollution exposure, potentially due to the capacity of EPA and DHA to stimulate antioxidant pathways and improve autonomic function.

## 4.2. Vitamin C and vitamin E

Antioxidants vitamin C and vitamin E have been the subject of numerous investigations because of their low cost and widespread use as dietary supplements around the world. Based on the findings of our study, two studies that investigated the effects of vitamin C supplementation on SBP in participants exposed to fine particle air pollution had conflicting results, one showing effects in lowering SBP, while the other did not show any benefits [23, 42]. The second antioxidant combination, which included vitamin C combined with NAC, was only tested in one trial and failed to show protective effects in vascular outcomes (*i.e.*, FMD) [44]. In the third case, co-supplementation with vitamins C and E proved to show protective effects against pulmonary function detriments in asthmatic adults and children exposed to $O_3$ [26, 28], and a combination of these vitamins with vegetable also showed protective effects against $O_3$ exposure [41]. Collectively, the available results do not offer much support for the efficacy of antioxidant vitamin supplementation in mitigating cardiovascular effects caused by air pollution; however, these supplements might offer protective effects against pulmonary damage induced by pollutants. Generally, vitamin C and vitamin E, as antioxidants found in the fluid covering the respiratory system, exert a potent antioxidant impact by neutralizing free radicals and preventing lipid peroxidation, respectively [55, 56]. Some *in vitro* research suggest that vitamin C may have antioxidant and antiviral properties when applied directly to airway cells [57, 58]. Also, vitamin E administration has been demonstrated to diminish $O_3$-induced cell death in fibroblasts and prevent allergen-induced NF E2-related factor 2 (NRF2) inhibition in asthmatic alveolar macrophages. This has led to the hypothesis that getting more antioxidants could help protect against the oxidative damage caused by breathing in polluted air [59–61].

On the other hand, the putative hypotensive impact of vitamin C, as seen in one study [42], may be attributed to its capacity to enhance the synthesis and bioavailability of nitric oxide (NO). Vitamin C exhibits the ability to scavenge superoxide radicals, hence diminishing their reactivity with NO and impeding the generation of peroxynitrite [62]. Nevertheless, the absence of cardiovascular protection by antioxidants in the face of air pollution has also been linked to the likelihood that the vasomotor reaction to air pollution relies less on oxidative stress and more on the activation of the autonomic nervous system [43]. Moreover, some evidence suggests that antioxidant supplementation in healthy individuals who do not have a deficiency in vitamin C or other conditions that lead to functional depletion may actually have a pro-oxidant effect [63, 64]. Other studies have suggested that ascorbic acid can inhibit flow and agonist-mediated vasodilation by blocking the release of endothelial-derived hyperpolarizing factors, leading to vasoconstriction [43, 65]. Considering the different dosing and duration of vitamin C supplementation, as well as different pollutant types, the methodological inconsistencies are another source of variations in the observed outcomes. Therefore, the inconsistencies in the cardiovascular benefits of antioxidants against air pollution warrant additional mechanistic and clinical investigations.

## 4.3. Sulforaphane

There is a lack of consistency in the data available on sulforaphane's efficacy in preventing airway inflammation, with one study offering protection against exhaust particles [33], and the other reporting no effects against $O_3$ exposure [36]. Possible reasons for these conflicting outcomes may involve inconsistent dosage, dosage forms, target populations, duration of consumption, and type of pollution [33, 36]. Cruciferous vegetables like broccoli contain naturally occurring isothiocyanate sulforaphane, which has anti-inflammatory and antioxidant properties by stimulating the transcription factor NRF2. Multiple *in vitro* investigations have reported the upregulation of NRF2 genes following treatment with sulforaphane [66, 67]. However,

information regarding its ability to stimulate the expression of antioxidant genes and shield against airway inflammation is inconclusive. Further research is required in this particular area.

## 4.4. Other diets

We identified several other dietary supplementations assessed against cardiopulmonary outcomes of air pollution exposure. One study demonstrated that taking L-arginine was proved to be safe and could effectively lower BP in hypertensive individuals while they walked outside under traffic-related air pollution [38]. These effects could be attributed to the roles of NO. Accordingly, NO, which is a mediator of vasodilation, is synthesized from L-Arginine and triggers the activation of the enzyme guanylyl cyclase. This enzyme further facilitates the guanylyl triphosphate conversion into cyclic guanosine monophosphate (cGMP). This will further induce smooth muscle relaxation, leading to a subsequent reduction in blood pressure.

Another study found that NAC protects against increasing airway responsiveness caused by PM2.5 inhalation and lowers the requirement for supplement bronchodilators in people who already have hyperresponsive airways [44]. The primary mechanism underlying the antioxidant properties of NAC is attributed to the capacity of its free thiol group to engage in chemical reactions with ROS and nitrogen species. The main function of NAC is attributed to its capacity to elevate the intracellular levels of glutathione (GSH), which plays a pivotal role in maintaining cellular redox stability. NAC possesses anti-inflammatory properties, which enable it to effectively diminish the expression of tumor necrosis factor-alpha (TNF-α) as well as interleukins (IL-6 and IL-1β). This reduction is achieved through the suppression of NF-κB activity. Despite numerous *in vivo* and *ex vivo* investigations demonstrating the significant biological effects of NAC and its potential therapeutic benefits, the efficacy of NAC in clinical research for various pathological disorders remains controversial [68–70].

Recent investigations have indicated that B vitamins (50mg/d vitamin B6, 1mg/d vitamin B12, and 2.5mg/d folic acid) may possess a specific protective effect against the cardiovascular consequences of PM2.5 exposure [27]. This protective effect is believed to be mediated through the modulation of epigenetic and inflammatory signaling pathways, which serve as the links between air pollution exposure, intermediate biomarkers, and cardiovascular outcomes. The administration of vitamin B6 and folic acid has been found to decrease the release of chemokines from peripheral blood mononuclear cells and reduce the pro-inflammatory chemicals in circulation. B vitamins are also crucial elements for the biochemical procedure of DNA methylation [71–73].

Our study had several limitations. In contrast to the prolonged and recurrent duration of real-life air pollution exposure, investigations have primarily concentrated on examining the physiological responses to short-term exposure. Furthermore, the possible confounding effects of various pollutants in different geographical locations, genetic backgrounds, socioeconomic characteristics, dietary disparities, supplement compliance, and daily exercise routines among participants have not been evaluated. Another limitation is that the findings obtained from a limited sample of healthy individuals may not accurately reflect the physiological alterations observed in groups that are more vulnerable to the impacts of pollution, including populations such as children, elderly adults, persons with a history of cardiovascular illnesses, and individuals living in extremely polluted areas. Also, The impact of air pollution on cardiorespiratory parameters was not observed in several investigations, thus preventing a proper assessment of the supplement's capacity to modulate this effect [25, 37, 39]. Additionally, the reviewed trials evaluated a wide range of supplement doses and durations, as well as diverse air pollutants with varying concentrations and exposure conditions. Therefore, efforts to standardize

outcome measures and interventions in future research would be valuable in facilitating comparability across studies and enhancing the robustness of evidence in this field. We also emphasize the need for well-designed randomized controlled trials with larger sample sizes to provide more definitive answers regarding the efficacy and optimal dosages of dietary supplements in mitigating the cardiopulmonary effects of air pollution. Moreover, the number of studies in the assessment of some supplements, including B vitamins, L-arginine, and NAC was limited according to our findings, which necessitates further trials to assess the efficacy of these supplements against air pollution toxicity.

## 5. Conclusion

Supplements containing anti-inflammatory and antioxidant agents show contradictory results against air pollution toxicity. It is unclear whether these inconsistencies are due to an insufficient dose, poor supplement formulation, poor timing, or an insufficient study population. As a low-cost and low-effort preventative intervention for reducing the disease burden linked with air pollution, increasing the antioxidant intake of communities through antioxidant-rich meals might be beneficial; however, additional trials are required to provide a clearer understanding of the effectiveness of these supplements against pollution toxicity.

## Supporting information

**S1 Checklist. PRISMA checklist.**
(DOCX)

**S1 Table. Search keywords and queries.**
(DOCX)

## Acknowledgments

This study received grant supported by the Research Department of the School of Medicine, Shahid Beheshti University of Medical Sciences, Tehran, Iran (Grant number: 43007533).

## Author Contributions

**Conceptualization:** Moein Zangiabadian.

**Methodology:** Moein Zangiabadian.

**Supervision:** Mohammad Javad Nasiri.

**Validation:** Moein Zangiabadian.

**Visualization:** Moein Zangiabadian.

**Writing – original draft:** Mehran Ilaghi, Fatemeh Kafi, Mohadeseh Shafiei.

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
