## [Decision Letter · Decision Letter 0]

6 Mar 2024

PONE-D-24-03107Dietary Supplementations to Mitigate the Cardiopulmonary Effects of Air Pollution Toxicity: A Systematic Review of Clinical TrialsPLOS ONE

Dear Dr. Zangiabadian,

Thank you for submitting your manuscript to PLOS ONE. After careful consideration, we feel that it has merit but does not fully meet PLOS ONE’s publication criteria as it currently stands. Therefore, we invite you to submit a revised version of the manuscript that addresses the points raised during the review process.

We look forward to receiving your revised manuscript.

Kind regards,

Rami Salim Najjar, Ph.D.

Academic Editor

PLOS ONE

Journal Requirements:

3. Please include your tables as part of your main manuscript and remove the individual files. Please note that supplementary tables (should remain/ be uploaded) as separate "supporting information" files.

Reviewers' comments:

Reviewer's Responses to Questions

**Comments to the Author**

1. Is the manuscript technically sound, and do the data support the conclusions?

Reviewer #1: Partly

Reviewer #2: Yes

2. Has the statistical analysis been performed appropriately and rigorously? 

Reviewer #1: N/A

Reviewer #2: N/A

3. Have the authors made all data underlying the findings in their manuscript fully available?

Reviewer #1: Yes

Reviewer #2: Yes

4. Is the manuscript presented in an intelligible fashion and written in standard English?

Reviewer #1: Yes

Reviewer #2: Yes

5. Review Comments to the Author

Reviewer #1: This paper attempts to inform the literature of the beneficial effects of certain dietary supplements in populations and settings of poor air quality/air pollutant exposure. Authors utilized a systematic review approach to assess the impacts of particulate matter and ozone exposure on cardiovascular and pulmonary functioning and the potential for dietary supplements to mitigate these effects. Results of this systemic review were inconclusive in regard to the overall benefit of dietary supplementation in these exposure scenarios. The inconclusive results may be due to the wide array of studies included in this review, as discussed in the comments below. In addition to stylistic and organizational shortfalls of the paper, there are deficiencies in the background and rationale provided as well as fundamental flaws in the review.

Comments:

1. This review is focused on the cardiopulmonary effects of air pollution. However, the introduction only discusses the basic mechanisms through which particulates and other pollutants contribute to pulmonary detriments with no mention of the cardiovascular effects. Discussion of how pollutant exposure contributes to cardiovascular disease and alterations in cardiovascular functioning (specifically those discussed in the review) should be included.

2. Authors do not provide context as to why these outcome measures were chosen for evaluation and how they are illustrative cardiopulmonary dysfunction as mentioned above.

3. Authors should also elaborate further as to how dietary supplements, drawing on epidemiological and mechanistic research, may mitigate the deleterious effects of pollutants.

4. Check for proper use of defined abbreviations throughout.

5. There are multiple B vitamins, authors should be more specific about which B vitamin or combination of them is being evaluated in each study.

6. Authors should mention dosages and duration of both supplementation and pollutant exposure within the text.

7. The discussion provides description of metabolic and cellular changes resulting from dietary supplementation and pollution but there is very little of the results into this discussion.

8. The inconclusiveness of these findings does not inform the literature. Inconclusiveness is likely related to the wide array of outcomes measures and the number of dietary supplement interventions assessed in the review. Due to the variety in the studies included, in the review, it is difficult to make comparisons or conclusions on specific dietary supplements or parameters of interest.

Reviewer #2: This systematic review on the potential for dietary supplements to prevent the cardiorespiratory effects of air pollution is very timely. The concerning health effects of air pollution are more prominent than ever, yet efforts to improve air pollution progress remains slow and consequently the health burden very large. There have been suggestions in the field that dietary supplements or medicines could ameliorate the health effects of air pollution, but a clear picture in missing.

The systematic review follows PRISMA guidelines. The manuscript is clearly written.

I have a few suggestions that the authors should address:

1. In addition to subclinical cardiorespiratory outcomes the review could have included hospitalisation for cardiorespiratory conditions and cardiorespiratory mortality. If this is not done, the authors should justify why not. Some other subclinical parameters could have been included such as carotid-intimal media thickness or coronary artery calcium score.

2. The authors could go further in their scrutiny of the results to look for emerging patterns. For example, could some of the inconsistency be explained by the differences in doses of the supplements or the duration of use, or the type and concentrations of pollutant? The authors raise these possibilities very briefly, but do not indicate whether the studies they have assessed provide insight on these.

3. Although animal studies are not within the scope of the review, the authors could mention some key animal studies with use of natural antioxidant/anti-inflammatory agents on air pollution to bolster their Discussion on where effects may or may not be present in the human studies.

4. In the Discussion it should be noted that in several studies the air pollution itself did not affect the cardiorespiratory parameters, therefore, the ability of the supplement to modify the effect of air pollution could not be appropriately assessed.

5. There are a few reviews on the ability of medicinal agents to ameliorate the effects of air pollution on the respiratory and cardiovascular systems (e.g. Romieu et al. Eur Respir J 2008; Tong. Biochim Biophys Acta 2016; Barthelemy et al. Int J Environ Res Public Health 2020; Miller, Pharmacol Ther 2022; please also check for others). These also encompass dietary supplements and there is an opportunity to describe whether the conclusions of these reviews align with the present study, and how the literature has developed since the most recent review.

6. In Table 4, please add a column to state the key findings of each study (ideally with risk estimate/change and a confidence interval/error estimate).

Minor points:

7. Third paragraph of intro – please mention other non-combustion sources of PM2.5 such as secondary aerosols. Noe that particle size also affects toxicity.

8. Second line of page 16 – Did the olive oil decrease the FMD or the effect of the pollution on FMD?

9. Page 17. Given the role of oxidative stress on NO, the authors may wish to speculate on why vitamin C did not reverse the cardiovascular effects of air pollution.

6. PLOS authors have the option to publish the peer review history of their article (what does this mean?). If published, this will include your full peer review and any attached files.

Reviewer #1: No

Reviewer #2: **Yes: **Mark R Miller

---

## [Author Response · Author response to Decision Letter 0]

15 Apr 2024

Date: 04/15/2024

Ref: PONE-D-24-03107

Revision of manuscript: " Dietary Supplementations to Mitigate the Cardiopulmonary Effects of Air Pollution Toxicity: A Systematic Review of Clinical Trials "

Dear Editor,

We are grateful for the invaluable comments you and the reviewers provided regarding our submitted manuscript.

We have carefully considered each of the comments and made appropriate modifications to the manuscript. For clarity, the changes made to the manuscript are evident through Track Changes, and we have provided a point-by-point response to each comment below.

We hope that the changes made to the manuscript address the concerns raised. We eagerly anticipate your response and acknowledge the dedication and expertise demonstrated by you and the reviewers throughout the peer-review process.

Yours sincerely,

Moein Zangiabadian

Moein Zangiabadian, MD, MPH, MHPE 

Endocrinology and Metabolism Research Center, Institute of Basic and Clinical Physiology Sciences, Kerman University of Medical Sciences, Kerman, Iran 

Email: zangiabadian1998@gmail.com

 

Reviewer 1

Reviewer's general comment: This paper attempts to inform the literature of the beneficial effects of certain dietary supplements in populations and settings of poor air quality/air pollutant exposure. Authors utilized a systematic review approach to assess the impacts of particulate matter and ozone exposure on cardiovascular and pulmonary functioning and the potential for dietary supplements to mitigate these effects. Results of this systemic review were inconclusive in regard to the overall benefit of dietary supplementation in these exposure scenarios. The inconclusive results may be due to the wide array of studies included in this review, as discussed in the comments below. In addition to stylistic and organizational shortfalls of the paper, there are deficiencies in the background and rationale provided as well as fundamental flaws in the review.

Response to general comment: The authors sincerely thank the reviewer for his/her positive attitude toward our manuscript and for providing insightful comments. We carefully considered the comments and made point-by-point adjustments, which are addressed below.

Reviewer's comment #1: This review is focused on the cardiopulmonary effects of air pollution. However, the introduction only discusses the basic mechanisms through which particulates and other pollutants contribute to pulmonary detriments with no mention of the cardiovascular effects. Discussion of how pollutant exposure contributes to cardiovascular disease and alterations in cardiovascular functioning (specifically those discussed in the review) should be included.

Response to comment #1: We appreciate the reviewer’s meticulous evaluation of our manuscript. In response, the introduction has been updated with additional evidence to support the cardiovascular effects of particulate matter. “In addition, there is mounting evidence from both clinical and epidemiological studies linking air pollution to cardiovascular disease. Several negative health effects, including hypertension, heart disease, stroke, and high blood pressure, are closely linked with PM2.5 and PM10 air pollution levels. A study found that for every 10.5 μg/m3 of PM2.5, the risk of ischemic heart disease, heart failure, arrhythmias, and cardiac arrest increases by 8~18%. This is likely due to mechanisms such as systemic inflammation, accelerated atherosclerosis, and affected cardiac autonomic function.”

Reviewer's comment #2: Authors do not provide context as to why these outcome measures were chosen for evaluation and how they are illustrative cardiopulmonary dysfunction as mentioned above.

Response to comment #2: We thank the reviewer for bringing up this concern. Generally, selecting the outcomes was based on an a priori approach to comprehensively review the literature on clinical trials. The selected outcomes have been prospectively submitted to our study protocol, available on the PROSPERO website. Additionally, the rationale for choosing the outcomes was based on their ability to capture the cardiopulmonary effects of air pollution exposure as well as the existence of sufficient clinical trials for a systematic review. The included measures are well-established markers of cardiovascular and respiratory health that can be impacted by the toxicity of air pollutants.

For the cardiovascular outcomes, blood pressure, heart rate, and heart rate variability provide insight into autonomic nervous system functioning and cardiac hemodynamics. Alterations in these parameters have been linked to an increased risk of adverse cardiovascular events. Additionally, measures of vascular function, such as brachial artery diameter and flow-mediated dilation, can indicate endothelial dysfunction, which is an early marker of cardiovascular disease development. Finally, changes in blood lipid profiles are associated with the development of atherosclerosis, a key pathological process underlying many cardiovascular diseases.

On the respiratory side, spirometric indices like forced expiratory volume and flow rates reflect lung function. Disruption of these measures can indicate pulmonary dysfunction and increased susceptibility to respiratory diseases. Additionally, Markers of airway inflammation, such as cell counts in induced sputum or bronchoalveolar lavage, provide a direct assessment of the inflammatory response within the lungs triggered by air pollution exposure. Together, these respiratory outcomes can identify the detrimental effects of air pollutants on the structure and function of the airways and lungs.

For a more detailed explanation and clarification, we added more details to the introduction section.

Reviewer's comment #3: Authors should also elaborate further as to how dietary supplements, drawing on epidemiological and mechanistic research, may mitigate the deleterious effects of pollutants.

Response to comment #3: The authors sincerely thank the reviewer for pointing this out. Indeed, in the discussion section we have provided detailed explanations of the potential mechanisms through which dietary supplements impact the cardiopulmonary outcomes of air pollution toxicity. A comprehensive, evidence-based approach is implemented for each dietary supplementation in its own subsection in the discussion section. However, in response to the reviewer’s suggestion, we also added an explanation to the introduction section.

Reviewer's comment #4: Check for proper use of defined abbreviations throughout.

Response to comment #4: We appreciate the reviewer’s concern. The whole manuscript was rechecked for the proper use of abbreviations and necessary changes were made. All abbreviations were defined in their first appearance in the text.

Reviewer's comment #5: There are multiple B vitamins, authors should be more specific about which B vitamin or combination of them is being evaluated in each study. 

Response to comment #5: We had included a single clinical trial utilizing vitamin B as an intervention. As previously indicated in Table 2, the vitamin B evaluated in the study was a vitamin B supplement containing B6, B12, and folic acid. We clarified this throughout the results section. Moreover, in the discussion section, all studies already point to specific vitamin B types.

Reviewer's comment #6: Authors should mention dosages and duration of both supplementation and pollutant exposure within the text.

Response to comment #6: In response to the reviewer’s suggestion, we added dosages and duration details to the text where necessary. Moreover, the dosage and duration/frequency of both pollutant and supplementation are also provided in detail in Table 4, 

Reviewer's comment #7: The discussion provides description of metabolic and cellular changes resulting from dietary supplementation and pollution but there is very little of the results into this discussion.

Response to comment #7: We thank the reviewer for bringing up this concern. Throughout the updated discussion section, we first emphasized the findings pertaining to each intervention according to our systematic review and then provided a discussion that now ties the proposed biological mechanisms to the findings observed in the included studies or addressed inconsistencies. By more tightly integrating the specific systematic review findings into the mechanistic explanations, we have strengthened the discussion to provide readers with a more cohesive synthesis of the current evidence base. We hope this addresses the reviewer's feedback and improves the overall quality and impact of the paper.

Reviewer's comment #8: The inconclusiveness of these findings does not inform the literature. Inconclusiveness is likely related to the wide array of outcomes measures and the number of dietary supplement interventions assessed in the review. Due to the variety in the studies included, in the review, it is difficult to make comparisons or conclusions on specific dietary supplements or parameters of interest. 

Response to comment #8: We appreciate the reviewer’s thoughtful feedback. Our review aimed to comprehensively evaluate the existing literature on dietary supplements and their potential role in mitigating the adverse effects of air pollution on cardiovascular and pulmonary outcomes. By including a broad range of studies, we aimed to provide a detailed understanding of the current state of evidence and to identify gaps in knowledge that warrant further research. However, to enhance the readability, we have organized the results based on each specific outcome, and in each outcome, we have provided the results of each intervention distinctively. Therefore, we believe that the number of outcomes and supplements is essential to the comprehensive systematic review of clinical trials (considering the few trials on this specific subject so far). 

Nevertheless, a significant portion of the inconclusiveness can be attributed to variations in dosage and duration of intervention/exposure. Therefore, we agree that efforts to standardize outcome measures and interventions in future research would be valuable in facilitating comparability across studies and enhancing the robustness of evidence in this field. We also emphasize the need for well-designed randomized controlled trials with larger sample sizes to provide more definitive answers regarding the efficacy and optimal dosages of dietary supplements in mitigating the cardiopulmonary effects of air pollution. 

Overall, we believe that despite the challenges posed by the diversity of studies, a comprehensive review of all trials in this field for the first time provides valuable insights into the current state of knowledge and highlights opportunities for further research to address the complex relationship between dietary supplements and air pollution exposure.

 

Reviewer 2

Reviewer's general comment: This systematic review on the potential for dietary supplements to prevent the cardiorespiratory effects of air pollution is very timely. The concerning health effects of air pollution are more prominent than ever, yet efforts to improve air pollution progress remains slow and consequently the health burden very large. There have been suggestions in the field that dietary supplements or medicines could ameliorate the health effects of air pollution, but a clear picture in missing. The systematic review follows PRISMA guidelines. The manuscript is clearly written.

Response to general comment: We sincerely thank the reviewer for their positive attitude toward the manuscript and for the detailed evaluation of our study. We appreciate the valuable comments and suggestions of the reviewer to improve the quality of our work. The suggestions raised by the reviewer are addressed in detail below. 

Reviewer's comment #1: In addition to subclinical cardiorespiratory outcomes the review could have included hospitalisation for cardiorespiratory conditions and cardiorespiratory mortality. If this is not done, the authors should justify why not. Some other subclinical parameters could have been included such as carotid-intimal media thickness or coronary artery calcium score.

Response to comment #1: We appreciate the reviewer’s comment. Generally, selecting the outcomes was based on an a priori approach where we initially made a comprehensive review of the literature based on the existing clinical trials. We attempted to choose the clinical cardiopulmonary outcomes that have been addressed in at least two trials with dietary supplements as interventions. The selected outcomes have been finalized by the study team and have been prospectively submitted to our study protocol available on the PROSPERO website. 

In response to the reviewer’s suggestion, given that our focus was on clinical trials conducted under controlled outpatient settings, hospitalization and mortality were not applicable endpoints in the included studies. Moreover, while we acknowledge the potential relevance of additional parameters such as carotid-intimal media thickness or coronary artery calcium score, our decision to narrow our outcomes was based on the availability of clinical trial designs with relevant dietary supplementation intervention. Therefore, we adhered to our prospective protocol based on our initial review of the literature.

Reviewer's comment #2: The authors could go further in their scrutiny of the results to look for emerging patterns. For example, could some of the inconsistency be explained by the differences in doses of the supplements or the duration of use, or the type and concentrations of pollutant? The authors raise these possibilities very briefly, but do not indicate whether the studies they have assessed provide insight on these.

Response to comment #2: The reviewer raises a fair point. As we previously indicated, a significant portion of the inconclusiveness of the results can be attributed to variations in dosage and duration of intervention/exposure in different clinical trials. For enhanced readability, in the revised version of the manuscript, we have first explicitly incorporated information regarding dosing and pollutant type into the results section to improve readability and facilitate comparison of findings across studies. Secondly, during the discussion and limitation sections we emphasized the potential role of methodological inconsistencies in the variation seen in results. 

Reviewer's comment #3: Although animal studies are not within the scope of the review, the authors could mention some key animal studies with use of natural antioxidant/anti-inflammatory agents on air pollution to bolster their Discussion on where effects may or may not be present in the human studies.

Response to comment #3: This is indeed an interesting suggestion and we are grateful for that. In response, we have added some instances of animal studies, in vitro studies, or reviews based on preclinical data to the discussion section.

Reviewer's comment #4: In the Discussion it should be noted that in several studies the air pollution itself did not affect the cardiorespiratory parameters, therefore, the ability of the supplement to modify the effect of air pollution could not be appropriately assessed.

Response to comment #4: Yes, and we have pointed this out wherever indicating the results of the studies. In response to the reviewer’s suggestion, we also emphasized this in the limitations of our study.

Reviewer's comment #5: There are a few reviews on the ability of medicinal agents to ameliorate the effects of air pollution on the respiratory and cardiovascular systems (e.g. Romieu et al. Eur Respir J 2008; Tong. Biochim Biophys Acta 2016; Barthelemy et al. Int J Environ Res Public Health 2020; Miller, Pharmacol Ther 2022; please also check for others). These also encompass dietary supplements and there is an opportunity to describe whether the conclusions of these reviews align with the present study, and how the literature has developed since the most recent review.

Response to comment #5: We are grateful for these suggestions. Throughout our introduction and discussion, we utilized several reviews to add both additional mechanistic insights and the trends in clinical approaches against air pollution toxicity. We were particularl

---

## [Decision Letter · Decision Letter 1]

13 May 2024

Dietary Supplementations to Mitigate the Cardiopulmonary Effects of Air Pollution Toxicity: A Systematic Review of Clinical Trials

PONE-D-24-03107R1

Dear Dr. Zangiabadian,

We’re pleased to inform you that your manuscript has been judged scientifically suitable for publication and will be formally accepted for publication once it meets all outstanding technical requirements.

Kind regards,

Rami Salim Najjar, Ph.D.

Academic Editor

PLOS ONE

Additional Editor Comments (optional):

Reviewers' comments:

Reviewer's Responses to Questions

**Comments to the Author**

1. If the authors have adequately addressed your comments raised in a previous round of review and you feel that this manuscript is now acceptable for publication, you may indicate that here to bypass the “Comments to the Author” section, enter your conflict of interest statement in the “Confidential to Editor” section, and submit your "Accept" recommendation.

Reviewer #2: All comments have been addressed

2. Is the manuscript technically sound, and do the data support the conclusions?

Reviewer #2: Yes

3. Has the statistical analysis been performed appropriately and rigorously? 

Reviewer #2: N/A

4. Have the authors made all data underlying the findings in their manuscript fully available?

Reviewer #2: Yes

5. Is the manuscript presented in an intelligible fashion and written in standard English?

Reviewer #2: Yes

6. Review Comments to the Author

Reviewer #2: The authors have given good thought to all my comments and made appropriate changes to address them. Thank you.

As very minor points:

on page 3, the authors may wish to add construction work, secondary formation of PM (e.g. from reactions with ammonia and nitrate/sulphate) and non-exhaust emissions from traffic (brake, tyre and road wear) as major sources of non-combustion PM.

On page 4, other prominent mechanisms of the cardiovascular effects of air pollution include imbalance between vasodilation/vasoconstriction, and promotion of blood clotting.

7. PLOS authors have the option to publish the peer review history of their article (what does this mean?). If published, this will include your full peer review and any attached files.

Reviewer #2: **Yes: **Mark R Miller

---

## [Editor Report · Acceptance letter]

21 May 2024

PONE-D-24-03107R1 

PLOS ONE

Dear Dr. Zangiabadian, 

I'm pleased to inform you that your manuscript has been deemed suitable for publication in PLOS ONE. Congratulations! Your manuscript is now being handed over to our production team.

Kind regards, 

on behalf of

Dr. Rami Salim Najjar 

Academic Editor

PLOS ONE